# Abstractive Text Summarization for Icelandic

**Þór Sverrisson**
Department of Computer Science
University of Iceland
Iceland
`ths220@hi.is`

**Hafsteinn Einarsson**
Department of Computer Science
University of Iceland
Iceland
`hafsteinne@hi.is`

## Abstract

In this work, we studied methods for automatic abstractive summarization in a low-resource setting using Icelandic text, which is morphologically rich and has limited data compared to languages such as English. We collected and published the first publicly available abstractive summarization dataset for Icelandic and used it for training and evaluation of our models. We found that using multilingual pre-training in this setting led to improved performance, with the multilingual mT5 model consistently outperforming a similar model pre-trained from scratch on Icelandic text only. Additionally, we explored the use of machine translations for fine-tuning data augmentation and found that fine-tuning on the augmented data followed by fine-tuning on Icelandic data improved the results. This work highlights the importance of both high-quality training data and multilingual pre-training in achieving effective abstractive summarization in low-resource languages.

## 1 Introduction

The task of automatic text summarization has been gaining interest in recent years due to the increasing amount of available information and the need for well-written summaries that preserve key information while being coherent and flowing naturally. Two main approaches to automatic text summarization are extractive and abstractive methods. Extractive methods compose the summary out of copies of important sections from the original text, whereas abstractive methods rephrase and shorten the text similar to how a human would (Tas and Kiyani, 2017). The rise of Transformer models (Vaswani et al., 2017) in natural language processing (NLP) has led to great advances in the field, particularly in abstractive summarization (Zhang et al., 2020). However, these models often rely on a large amount of text data and computational resources for pre-training. This raises the question of whether low-resource languages can build advanced NLP models for summarization, given the lack of data.

We aim to address this question by studying the use of state-of-the-art Transformer models for abstractive summarization of Icelandic text. We introduce the first publicly available abstractive summarization dataset for Icelandic, RÚV Radio News (RRN), and use it for training and evaluation of the models. With that approach, we aim to study whether state-of-the-art Transformer models can be adapted to perform abstractive summarization in a low-resource setting for Icelandic text. In order to support future research on abstractive summarization in Icelandic, we are sharing our dataset[1] and the fine-tuned model[2] with the research community.

This work is motivated by the increasing demand for automatic text summarization and the challenges of applying machine learning methods to low-resource languages such as Icelandic. The study of NLP in low-resource languages is important for language preservation, and this research contributes to this field by providing a dataset for Icelandic and evaluating the performance of state-of-the-art Transformer models on it. Summarization has been claimed to be challenging in low-resource settings (Zoph et al., 2016; Khurana et al., 2022) and the potential solution that we base our work on is to apply transfer learning (Zhuang et al., 2021) and data augmentation techniques (Tanner and Wong, 1987).

---

[1] `https://huggingface.co/datasets/thors/RRN`
[2] `https://huggingface.co/thors/mt5-base-icelandic-summarization`

## 2 Background

Abstractive summarization is a complex task that involves identifying important information from a text and expressing it in new words. The Transformer architecture (Vaswani et al., 2017), which is based on the attention mechanism (Bahdanau et al., 2015), has become popular for this task as it can efficiently work with larger text segments and take into account context in the input.

Transformers are widely applied through transfer learning, a technique introduced by Yosinski et al. (2014) where a model trained on one task is fine-tuned or reused as the starting point for a model on a similar or different task. Prior to the transfer, the models are generally trained using self-supervision, which allows the models to leverage a large, diverse corpus of unlabeled text data. For generative models, the pre-training objective often involves masking parts of the input sequence and tasking the model with filling in the gaps, as proposed by (Song et al., 2019) for example. Raffel et al. (2020) demonstrated with the T5 model that many NLP problems can be treated as text-to-text tasks, allowing for the pre-training of a single encoder-decoder Transformer on a diverse set of tasks. Additionally, BART models (Lewis et al., 2020) have been trained to reconstruct a text document that has been corrupted with an arbitrary noising function and have proved to be very effective at tasks such as summarization. The PEGA-SUS model (Zhang et al., 2020) uses a pre-training objective that closely resembles the summarization task, resulting in a model that adapts faster when fine-tuned on a small number of examples.

Pre-training language models through self-supervised learning has achieved impressive results when applied to abstractive summarization tasks. However, obtaining high-quality summarization outcomes can be difficult when there is a scarcity of data for fine-tuning, a common issue encountered with low-resource languages. To tackle this challenge, researchers have turned to transfer learning and data augmentation techniques, which have proven to be effective in various low-resource natural language processing (NLP) tasks (Hedderich et al., 2021). Prior results on abstractive summarization in a low-resource setting serve as good examples of applying such methods (Fadaee et al., 2017; Sennrich et al., 2016).

Transfer learning methods have enabled progress in Icelandic NLP tasks, such as translation (Símonarson et al., 2021), question answering (Snæbjarnarson and Einarsson, 2022b), and named entity recognition (Snæbjarnarson et al., 2022). However, research on Icelandic summarization has predominantly concentrated on extractive approaches (Christiansen, 2014; Daðason et al., 2021; Daðason and Loftsson, 2022). Multilingual models, like XLM-R (Conneau et al., 2020) and mT5 (Xue et al., 2021), have exhibited promising results across a wide range of NLP tasks and have been particularly advantageous for Icelandic tasks (Snæbjarnarson et al., 2022; Snæbjarnarson and Einarsson, 2022a).

## 3 Methods

### 3.1 Data

A summary of the text corpora utilized in this study is provided in Table 1. The English language corpora were translated to Icelandic using machine translation, as described in Section 3.1.4.

### 3.1.1 Pre-training Corpus

The **Icelandic Gigaword Corpus** (IGC, (Steingrímsson et al., 2018)) version 20.05 was used for pre-training of the Gullfaxi model (see Section 3.2). The corpus consists of a collection of approximately 5 million documents from various categories, including adjudications, parliamentary speeches, news, books, and scientific journals. The corpus consists of text that is automatically divided into sentences and running words, tagged, and lemmatized. The IGC-News1 21.05 dataset, consisting of news articles from the year 2020, was used for validation during pre-training. These articles were not included in the training data.

### 3.1.2 Fine-tuning corpora

In this study, we utilized the following news summarization datasets for fine-tuning our models:

**RÚV Radio News** (RRN) dataset, which consists of news stories from the Icelandic National Broadcasting Service (RÚV) collected specifically for this study. It includes 4k stories from 2021 and 2022, containing many stories related to COVID-19 and domestic news.

**XSum** dataset (Narayan et al., 2018), which features a variety of English-language BBC articles from 2010 to 2017, each accompanied by a professional, single-sentence summary.

**CNN/DailyMail** dataset (Hermann et al., 2015), which includes English-language news stories

| Dataset | # Documents | Language | Type |
|---|---|---|---|
| IGC 20.05 | 5M | is | Generic |
| IGC-News1 21.05 (2020) | 112k | is | Generic |
| RRN | 4k | is | Summarization |
| XSum | 227k | en | Summarization |
| CNN/DailyMail | 311k | en | Summarization |

Table 1: Overview of the datasets used in this study. The language column refers to the original language of the dataset.

from CNN and Daily Mail websites, each accompanied by human-written summary bullets.

Note that there was no overlap between the fine-tuning datasets and the pre-training corpus. We study fine-tuning on the datasets separately and we also study fine-tuning on translated data followed by fine-tuning on RRN.

### 3.1.3 Pre-processing RRN

The Icelandic National Broadcasting Service (RÚV) granted access to a database of news stories via a custom interface that was available on-premises at their headquarters. The stories were manually selected from the database, and only transcripts of radio news from 2021 and 2022 were used. The RRN dataset was extracted from these transcripts and comprises four parts for each story: a title, an intro, the main story, and a summary. To ensure that the dataset was suitable for the summarization task, we filtered out stories that were not relevant, such as live broadcasts and weather news. Additionally, we programmatically removed reporters' comments, phone numbers, and instructions for the broadcast. The intro and the summary are often similar as they both provide an overview of the key points of the story and in some instances, they may be identical. For a given date, the summaries were in a separate document and not linked to a story by any unique identifier. Therefore, the summaries were paired with their corresponding stories in a heuristic manner using a ROUGE1-F1 score. To ensure the accuracy of the pairing, we reviewed 100 random pairings and found that this approach produced correct pairings in all cases.

### 3.1.4 English to Icelandic Translation

In order to augment our summarization data, we translated the XSum and CNN/DailyMail datasets from English to Icelandic using a machine translation model. Specifically, we utilized Facebook's multilingual model, which was a winning submission to the 2021 Conference on Machine Translation (WMT, Tran et al. (2021)). This model is fine-tuned for news domain data and trained using data from eight different languages, achieving state-of-the-art performance in machine translation. We used the pre-trained version of the model, which is available in HuggingFace's Transformers library, and loaded the weights from the `wmt21-dense-24-wide-en-x` repository. To improve the quality of translations, we split the text into sentences and translated them separately. This approach was found to improve translation quality during a manual inspection, although no quantitative evaluation was performed to confirm it.

### 3.2 Models

In this study, we introduce the Gullfaxi model, which is based on the PEGASUS architecture (Zhang et al., 2020) but trained on Icelandic text. We call the model Gullfaxi$_{BASE}$ and it corresponds to the BASE architecture presented in the PEGASUS study. Gullfaxi$_{BASE}$ has 223M trainable parameters. Additionally, we also fine-tune a pre-trained mT5 model (Xue et al., 2021) for performance comparison. We use mT5$_{BASE}$, which has 580M trainable parameters. The increase in parameter count compared to Gullfaxi $_{BASE}$ is primarily due to the larger vocabulary employed in mT5. Details on training and hyperparameters can be found in Appendix A

### 3.3 Downstream Tasks

We evaluate the performance of our models on a set of downstream summarization tasks using the RÚV Radio News (RRN) dataset. The RRN dataset is split into train, validation, and test sets with a 60%, 20%, 20% ratio respectively. We created three fine-tuning tasks to test different abilities for abstractive summarization:

**Task 1: Intro + Main → Summary** The task involves producing a summary from the introduction and main part of the story. As the introduction and summary are often similar and in some cases identical, this task is somewhat related to extractive summarization.

**Task 2: Main → Intro** The task involves generating the introduction from the main part of the story. The introduction and main text rarely share the same sentences, thus we expect the model to generate more abstractive summaries.

**Task 3: Intro → Title** The task involves producing the title of the story from the introduction. The title is much shorter compared to the output in the previous tasks, and we expect the model to generate more abstractive summaries.

To further understand the performance of Gullfaxi on larger corpora, we fine-tune it on the Icelandic translations of the XSum and CNN/DailyMail datasets and compare the results to the English PEGASUS model.

We also explored a mixed fine-tuning approach where the models were first fine-tuned on translated data and then on Icelandic data. For each fine-tuning phase, the model was fine-tuned until the validation loss stopped decreasing.

### 3.4 Performance Measures

In this study, we use the Recall-Oriented Understudy for Gisting Evaluation (ROUGE) scoring algorithm to evaluate the performance of our models (Lin, 2004). ROUGE is a widely used and accepted standard for evaluating automatic summarization tasks. We use ROUGE-1, ROUGE-2, and ROUGE-L to calculate the similarity between the model's summary and a reference summary.

We define $\text{count}_{\text{match}}(\text{gram}_n)$ as the number of matching n-grams, and similarly, $\text{count}_{\text{ref}}(\text{gram}_n)$ and $\text{count}_{\text{model}}(\text{gram}_n)$ refer to the number of n-grams in the reference and the model output, respectively. The ROUGE-N precision, recall, and F1-score are calculated as follows:

$$\text{ROUGE-N precision} = \frac{\text{count}_{\text{match}}(\text{gram}_n)}{\text{count}_{\text{model}}(\text{gram}_n)},$$

$$\text{ROUGE-N recall} = \frac{\text{count}_{\text{match}}(\text{gram}_n)}{\text{count}_{\text{ref}}(\text{gram}_n)}.$$

Similarly, we define ROUGE-L precision and recall using the longest common subsequence (LCS) between the reference summary and the model's output in the numerator. The LCS represents the longest sequence of words shared between the two texts, regardless of whether the words appear consecutively. Finally, we compute the F1-score for each of ROUGE-1, ROUGE-2, and ROUGE-L as the harmonic mean of their precision and recall.

$$\text{F1-score} = 2 \times \frac{\text{precision} \times \text{recall}}{\text{precision} + \text{recall}}.$$

### 3.5 Human Evaluation

To further assess the quality of the generated summaries, we conduct human evaluations on a subset of the generated summaries. The samples are rated by a single annotator on three binary criteria: relevance, correctness, and language. Relevance is based on whether the summary is relevant to the reference text and pertains to the subject matter of the story. Correctness is based on whether the summary is factually accurate and consistent with the reference text, and does not include any unrelated information. Lastly, language is based on whether the summary is grammatically correct and natural, without any repetitions or use of non-Icelandic words.

## 4 Results

### 4.1 Summarization Performance

In this section, we present the results of our evaluation of the Gullfaxi model and the mT5 model on the RRN dataset. Table 2 shows the ROUGE F1-scores (R1/R2/RL) of the fine-tuned models for each task. The results show that mT5$_{\text{BASE}}$ outperforms the Gullfaxi model on all tasks. The difference between the models is particularly notable in the first task (Intro + Main → Summary). Opting for an extractive approach in this task provides leverage in achieving high ROUGE scores as the intro and the summary tend to be similar. For comparison, a basic strategy of copying the intro yields a score of $61.8/46.2/58.9$. Examples of the model outputs and their scores can be found in Appendix B.

We also found that mT5 almost exclusively relied on an extractive approach in the first task, simply copying the intro, which resulted in a much higher score compared to Gullfaxi. In the other tasks, we observed more abstractive output from all models. Factors that contributed to lower ROUGE scores include repetition, grammatical errors, and different lengths of the output.

As a reference, we also fine-tuned a randomly initialized model, referred to as Transformer$_{BASE}$, with the same architecture as Gullfaxi$_{BASE}$ on the full RRN training set without any pre-training.

## 4.2 Low-resource Fine-tuning

In this section, we examine the performance of Gullfaxi and mT5 in a low-resource fine-tuning setting. We fine-tuned both models using varying amounts of data from the RRN dataset, specifically using the first $10^k$ ($k = 1, 2, 3$) examples from the training set. Figure 1 show the results of the low-resource fine-tuning of Gullfaxi$_{BASE}$ and mT5$_{BASE}$.

Our findings indicate that even without fine-tuning, Gullfaxi$_{BASE}$ performed better than Transformer$_{BASE}$ on some tasks. mT5$_{BASE}$ also showed a gradual improvement in performance as the number of training examples increased. Both models achieved significantly higher scores than Transformer$_{BASE}$ when fine-tuned on the full RRN training set.

## 4.3 Fine-tuning Data Augmentation

In this section, we investigate the impact of data augmentation on fine-tuning Gullfaxi and mT5 for summarization tasks. Specifically, we fine-tune the models on the Icelandic translations of XSum and CNN/DailyMail datasets and evaluate their performance on the RRN dataset. We also explore an approach where the model is fine-tuned in two phases, first on augmented data and then on RRN data. Results are presented in Table 2. We observe that when the translations are combined with RRN, the scores are higher. Furthermore, by manually reviewing the output of the models, we notice an increase in grammatical errors when using the translations for fine-tuning for Gullfaxi but not for the mT5 model. To further demonstrate the difference in performance between Icelandic and English models trained in a similar manner, we evaluate the performance of the Gullfaxi model fine-tuned on XSum and CNN/DailyMail on their respective test sets. Table 3 shows the results with and without fine-tuning, as well as a comparison to the English data scores of the PEGASUS models obtained from the original study. It is apparent that fine-tuning leads to a notable improvement in performance on both datasets. However, when comparing Gullfaxi to PEGASUS, it is evident that the PEGASUS model's scores for English are much higher.

## 4.4 Human Evaluation

In order to further evaluate the performance of our models, we conducted a human evaluation of a subset of the summary outputs. We randomly sampled 50 examples from the Main $\rightarrow$ Intro task, which tests the model's ability to generate an abstractive summary in a few sentences. The results of the human evaluation are presented in Table 4, which compares the scores for Gullfaxi and mT5 for different fine-tuning approaches.

In general, we observed that the output intros produced by all models were of lower quality than those written by humans. The outputs were often relevant to the reference text but not effectively summarizing it. Fine-tuning Gullfaxi on the Icelandic translations of CNN/DailyMail resulted in the worst performance, particularly regarding grammar, often using the wrong inflections of words, as seen in Table 5.

We further observed that the mT5 model improved with augmented translation data, whereas Gullfaxi performed worse with the augmented data, particularly in grammar and word inflection. Overall, the mT5 model showed superior performance, producing the best summaries when fine-tuned on the augmentations followed by fine-tuning on the RRN dataset, demonstrating generalization to the summarization task. However, it sometimes extracted information from the reference text instead of generating new phrases, which may explain its higher scores for relevance and correctness compared to Gullfaxi.

## 5 Discussion

In this study, we investigated techniques for addressing the challenging task of low-resource abstractive summarization for Icelandic. We evaluated several well-known approaches and uncovered limitations as well as potential avenues for future work.

The main challenge in our study was the lack of sufficient data for training abstractive summarization models for Icelandic. We collected a news-domain abstractive summarization dataset, RÚV Radio News (RRN), but acknowledge that it is relatively small and may not generalize well to other domains or summarization settings. The collection and processing of RRN were time-consuming due to the inconsistency in the format of the radio transcripts. To aid in future language resource development, publicly funded organizations, such as

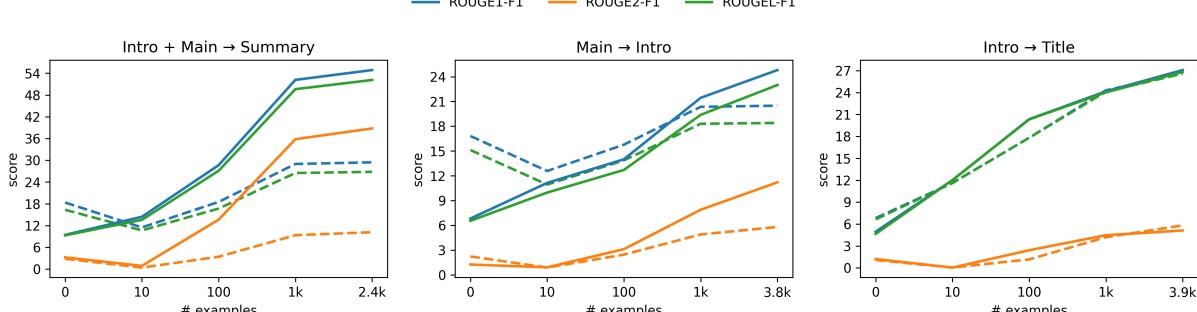

Figure 1: Model performance on RRN with a limited number of fine-tuning examples. The dashed lines are the performance of the Gullfaxi$_{\text{BASE}}$ model whereas the solid lines represent the mT5$_{\text{BASE}}$ model.

| Model | Intro + Main → Summary | Main → Intro | Intro → Title |
|---|---|---|---|
| Transformer$_{\text{BASE}}$ (only fine-tuning) | 13.1/1.3/12.0 | 10.8/0.7/9.7 | 14.2/0.6/14.1 |
| Gullfaxi$_{\text{BASE}}$ (no fine-tuning) | 18.4/2.9/16.4 | 16.8/2.2/15.1 | 6.8/1.1/6.6 |
| Gullfaxi$_{\text{BASE}}$ (RRN) | 29.4/10.2/26.8 | 20.5/5.7/18.5 | 26.6/5.8/26.6 |
| Gullfaxi$_{\text{BASE}}$ (CNN/DailyMail) | 26.5/9.0/24.2 | 17.8/3.5/16.0 | - |
| Gullfaxi$_{\text{BASE}}$ (CNN/DailyMail + RRN) | 42.5/21.3/39.6 | 22.2/6.2/19.7 | - |
| mT5$_{\text{BASE}}$ (RRN) | 54.9/38.8/52.1 | 24.8/11.2/23.0 | 27.1/5.1/26.8 |
| mT5$_{\text{BASE}}$ (CNN/DailyMail) | 36.2/19.3/33.9 | 21.4/5.9/19.4 | - |
| mT5$_{\text{BASE}}$ (CNN/DailyMail + RRN) | **58.9/42.8/56.1** | **33.0/17.0/30.6** | - |

Table 2: A comparison of Gullfaxi$_{\text{BASE}}$ and mT5$_{\text{BASE}}$ on the RRN dataset using different training sets. Transformer$_{\text{BASE}}$ has the same model architecture as Gullfaxi$_{\text{BASE}}$ but is not pre-trained, only randomly initialized. The scores listed are the ROUGE F1-scores (R1/R2/RL). The information in brackets denotes what data the model was fine-tuned on, when fine-tuned on more than a single dataset, the training is performed in two phases. Highest scores in the first two columns are shown in bold.

| Model | XSum$_{\text{is}}$ | CNN/DailyMail$_{\text{is}}$ |
|---|---|---|
| Gullfaxi$_{\text{BASE}}$ (no fine-tuning) | 13.3/1.1/11.4 | 13.1/1.3/12.1 |
| Gullfaxi$_{\text{BASE}}$ (XSum) | 23.5/7.3/19.9 | - |
| Gullfaxi$_{\text{BASE}}$ (CNN/DailyMail) | - | 24.6/7.7/23.1 |
| | **XSum$_{\text{en}}$** | **CNN/DailyMail$_{\text{en}}$** |
| PEGASUS$_{\text{BASE}}$ | 39.8/16.6/31.7 | 41.8/18.8/38.9 |

Table 3: Gullfaxi$_{\text{BASE}}$'s ROUGE F1-scores (R1/R2/RL) with and without fine-tuning on the Icelandic translations of XSum and CNN/DailyMail. The scores listed for PEGASUS$_{\text{BASE}}$ are the highest English language fine-tuning scores obtained from the PEGASUS paper.

RÚV, should be encouraged to be more mindful of their data processing. RRN provides a starting point for further research in this field, but broader coverage and diversity are necessary to create practical summarization solutions for Icelandic.

We evaluated the performance of two Gullfaxi models and mT5 on the RRN dataset for abstractive summarization in a low-resource setting, specifically for the Icelandic language. mT5 consistently outperformed the Gullfaxi models. However, we also observed that pre-training Gullfaxi led to better summarization performance when compared to no pre-training.

The performance of the multilingual mT5 model can be attributed to the large corpus of multilingual data, including 2.1 million Icelandic pages, used for pre-training. It should further

| Model | Relevance | Correctness | Language |
|---|---|---|---|
| Gullfaxi$_{\text{BASE}}$ (RRN) | 74% | 8% | 42% |
| Gullfaxi$_{\text{BASE}}$ (CNN/DailyMail) | 46% | 8% | 6% |
| Gullfaxi$_{\text{BASE}}$ (CNN/DailyMail + RRN) | 64% | 4% | 10% |
| mT5$_{\text{BASE}}$ (RRN) | 84% | 46% | 54% |
| mT5$_{\text{BASE}}$ (CNN/DailyMail) | 80% | 42% | 44% |
| mT5$_{\text{BASE}}$ (CNN/DailyMail + RRN) | **96**% | **54**% | **56**% |

Table 4: The human evaluation scores for the Main → Intro task. The scores listed are the fraction of summary results that fulfilled the criteria of each category in the 50 annotations evaluated. Highest scores in each column are shown in bold.

| | |
|---|---|
| **Reference summary** | Framkvæmdastjóri Vistorku á Akureyri segir raunsæjan kost að Ísland geti orðið algjörlega óháð olíu á næstu árum og þar með sjálfbært um alla orkuframleiðslu. Heildræna stefnu vanti þó í málaflokknum. |
| **Model output** | framkvæmdastjóri segir að það **er engin** (séu engar) hindranir **til** (fyrir því) að flýta banni við innflutningi á olíu |
| **Reference summary** | Í fyrsta sinn í 15 ár er stefnt að því að byggja fjölda íbúðarhúsa norður af Akureyri. Mikill áhugi er á lóðunum og færri fengu úthlutun en vildu. |
| **Model output** | á síðustu 19 árum **hafa lóðir** (hefur lóðum) verið úthlutað til eldri borgara. áhugi á **lóðir** (lóðum) hefur aukist á undanförnum árum. |

Table 5: Examples of ungrammatical output text of Gullfaxi$_{\text{BASE}}$ fine-tuned exclusively on the Icelandic translations of CNN/DailyMail. Corrections are in parentheses. The inflections of the words in red are incorrect.

benefit from the translation task, which is one of the tasks it is trained on in the pre-training phase. Our results suggest that low-resource languages may benefit from the general knowledge acquired through multilingual pre-training when fine-tuned for specific tasks, aligning with previous work (Snæbjarnarson et al., 2022; Snæbjarnarson and Einarsson, 2022a) where multilingual models for Icelandic were studied. For Gullfaxi, we used the same hyperparameters as in the Pegasus paper, but we still cannot conclude that Gullfaxi cannot be made better since we did not perform extensive hyperparameter tuning of the model due to time and cost.

We explored using machine translations to augment data for low-resource NLP tasks, specifically abstractive summarization in Icelandic. We fine-tuned models on Icelandic translations of two large English summarization datasets, CNN/DailyMail and XSum, but found the fine-tuned model did not perform well on the Icelandic summarization task, RÚV Radio News (RRN) and had more grammar mistakes compared to other models. When reviewing the Icelandic translations used for data augmentation there are a few things to note. Although most of them are easily comprehensible for a native speaker, they tend to be unnatural, use unusual wording, and have the wrong inflection of words. For that reason, we think that exclusively using translated examples for fine-tuning can sometimes lead to worse output texts.

We also explored a two-phase fine-tuning approach where we first fine-tuned on translated data and then on RRN. We observed improvements in ROUGE metrics but a manual inspection revealed better summaries for the mT5 model but worse summaries for the Gullfaxi model when compared to using no augmentation. This difference highlights the limitation of using ROUGE scores as a metric to measure summarization performance. It further highlights the importance of the quality of training data in low-resource settings, as well as the importance of considering the naturalness and grammatical accuracy of machine translations when using them for data augmentation.

We investigated the impact of the number of fine-tuning examples on the performance of a low-resource abstractive summarization task by fine-tuning Gullfaxi$_{BASE}$ with 0, 10, 100, and 1k examples from the RRN dataset. Our results showed that even without fine-tuning, Gullfaxi$_{BASE}$ performed better on some tasks than a randomly initialized model fine-tuned on all the RRN training set examples. As we increased the number of fine-tuning examples, Gullfaxi$_{BASE}$ continued to improve, achieving significantly higher scores than the baseline when using the full RRN dataset. This demonstrates the effectiveness of pre-training in a low-resource setting and highlights the potential value of creating small, domain-specific summarization datasets. However, we also observed that the mT5 model was better able to make use of more fine-tuning examples when exceeding a thousand examples.

The study has several limitations, including that all models tend to generate summaries that are inconsistent with the source text, which is a common issue for abstractive summarization models, and limits their practical use (Cao et al., 2018; Bender et al., 2021). Another limitation is that the pre-training objective may encourage the generation of incorrect statements. To address this, the use of reinforcement learning with human feedback, as demonstrated by the Instruct GPT model (Ouyang et al., 2022) can be used. Additionally, it is worth noting that most state-of-the-art models and breakthrough studies in NLP are primarily focused on English-language solutions, and it is unclear to what extent the choice of language impacts the performance of these models when the training budget and amount of training data are fixed. Further research comparing the performance of state-of-the-art models across different languages would be necessary to better understand this issue. Lastly, we would like to highlight the potential of including the Main → Summary task in future research, which was deemed out of scope for this work.

Our evaluation approach may be perceived as a limitation due to its binary nature. However, we intentionally designed it this way to prioritize objectivity, by being stringent about any errors in the model-generated summaries. That said, there are alternative evaluation approaches that could be explored in future research, such as employing continuous rating scales or more nuanced assessment criteria to better capture the intricacies of summary quality. By investigating these alternatives, we can potentially gain a deeper understanding of the strengths and weaknesses of summarization models.

Our results on data augmentation show that evaluating abstractive summaries is challenging. In this study, we used ROUGE scores and human evaluation, but ROUGE has been known to favor lexical similarity, which may not be suitable for abstractive summaries (Ng and Abrecht, 2015), particularly in morphologically rich languages like Icelandic. The lower ROUGE scores of Icelandic summaries compared to English language studies may be due to the differences in grammar between the two languages. The human evaluation revealed that the summaries tended to be factually inaccurate and had varying levels of grammatical quality. For future evaluations, it could be beneficial to include human-written summaries for comparison.

## 6 Conclusion

In this work, we explored methods for automatic abstractive summarization in a low-resource setting, specifically in the Icelandic language. We collected and published the first publicly available abstractive summarization dataset for Icelandic, and used it to train and evaluate state-of-the-art models. Our findings indicate that multilingual pre-training provides significant benefits for this task, as the multilingual mT5 model consistently outperformed a similar capacity PEGASUS model pre-trained from scratch on Icelandic text only. Additionally, we found that using machine translations for data augmentation led to higher ROUGE scores. However, when evaluated manually, the benefits of data augmentation were not consistently observed across models compared to a scenario where models were solely fine-tuned on the RRN dataset. Specifically, data augmentation enhanced the quality of the summaries generated by the mT5 model compared to those produced with RRN fine-tuning alone. In contrast, the Gullfaxi model's summaries experienced a decrease in quality due to data augmentation, displaying weaker grammar and a higher level of inconsistency compared to the reference text.

For future work, we suggest a further collection of abstractive summarization data for Icelandic, as well as studying metrics that may be better suited for this language. We also emphasize the benefits

of using pre-trained multilingual models, which we expect to apply to other generative tasks and languages. Overall, our study highlights the importance of pre-training and the challenges of evaluating abstractive summarization in low-resource settings.

## Acknowledgments

We thank Prof. Dr.-Ing. Morris Riedel and his team for providing access to the DEEP supercomputer at Forschungszentrum Jülich.

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

# A  Training details

## A.1  Pre-training Objective

Gullfaxi is pre-trained using a self-supervised pre-training objective called gap sentence generation

(GSG). This method, originally proposed for the PEGASUS model, involves masking whole sentences from the input document and concatenating them in their original order to form an abstractive summary-like output text. The goal is that a pre-training objective that closely resembles the summarization task will lead to a better starting point for fine-tuning.

The gap sentences are selected using a heuristic approach based on importance criteria. The ideal outcome is that the sentences containing the key information of the text are chosen from the document, but this is not guaranteed by the method. The importance of a sentence within a document is estimated by calculating the ROUGE1-F1 score between the gap sentence and the rest of the document. In this study, we calculate this score based on the lemmatized sentences as they are given in the IGC, due to the inflected nature of the Icelandic language.

The highest-performing models from the PEGASUS study were obtained by choosing a gap sentence ratio between 15%-45%, varying by task and model. For this study, we mask 20% of the total number of sentences in the original text document.

### A.2 Vocabulary

The vocabulary of a language model is the set of unique subword units, referred to as tokens, that the model is able to recognize. Methods such as PEGASUS and mT5 construct the vocabulary by training a subword tokenizer, which aims to identify an appropriate separation of input text. In this study, we use the SentencePiece unigram tokenizer (Kudo and Richardson, 2018) to construct a vocabulary for Gullfaxi. This configuration is similar to that used in PEGASUS, with a vocabulary size of 96k and no differentiation between lowercase and uppercase letters. The Gullfaxi tokenizer is trained on documents from the Icelandic Gigaword Corpus (IGC). On the other hand, the mT5 model comes with a pre-trained multilingual vocabulary of size 250k, obtained from training a SentencePiece tokenizer on the mC4 dataset.

### A.3 Hyperparameter configuration

In this study, we use the same hyperparameter configuration as the PEGASUS model, as it is computationally expensive to train and conduct a search for optimal hyperparameters. Details of the experiments' hyperparameters and training configuration can be found in the appendix. The Gullfaxi model is trained from scratch and implemented using the HuggingFace Transformers library, while the mT5 model uses pre-trained weights from the `google/mt5-base` repository and the corresponding tokenizer. The training was conducted on a high-performance computing cluster using multiple GPUs and distributed configuration with the HuggingFace Accelerate library. During fine-tuning, we use a label-smoothed regularization with a value of 0.1, and at test time, we use a beam size of 8 with a length penalty of 0.8 for all tasks.

## B  Example model outputs

Examples of model output can be seen in Tables 7, 8, and 9.

| Pre-training of Gullfaxi | | | | |
|---|---|---|---|---|
| **Model** | **# Steps** | **Batch size** | **Max input tokens** | **Max target tokens** |
| Gullfaxi$_{BASE}$ | 200k | 256 | 512 | 256 |
| **Fine-tuning of Gullfaxi models in Table 2, 3 and 4** | | | | |
| **Task** | **# Steps** | **Batch size** | **Max input tokens** | **Max target tokens** |
| Intro + Main $\rightarrow$ Summary | 4k | 256 | 512 | 128 |
| Main $\rightarrow$ Intro | 4k | 256 | 512 | 128 |
| Intro $\rightarrow$ Title | 4k | 256 | 128 | 32 |
| XSum | 50k | 256 | 512 | 64 |
| CNN/DailyMail | 50k | 256 | 512 | 128 |
| **Fine-tuning of mT5$_{BASE}$ in Table 2 and 4** | | | | |
| **Task** | **# Steps** | **Batch size** | **Max input tokens** | **Max target tokens** |
| RRN | 8k | 256 | Same as Gullfaxi | |
| **Low-resource fine-tuning of Gullfaxi$_{BASE}$ and mT5$_{BASE}$ in Figure 1** | | | | |
| **Task** | **# Steps** | **Batch size** | **Max input tokens** | **Max target tokens** |
| RRN | 3k | 256 | Same as Gullfaxi | |

Table 6: Hyperparameter setup for pre-training and fine-tuning.

| RRN Document | |
|---|---|
| **Title** | Neyðarástand vegna flóða í Kína (e. Emergency due to floods in China) |
| **Intro** | Hæsta viðbúnaðarstigi hefur verið lýst yfir í Henan-héraði í Kína vegna flóða. Þau hafa orðið að minnsta kosti tólf manns að bana. (e. The highest level of preparedness has been declared in Henan province in China due to floods. At least twelve people have been killed. |
| **Main** | Hátt í tvö hundruð þúsund íbúar borgarinnar Sheng-sjá Zhengzhou í Henan-héraði í Kína hafa verið fluttir að heiman vegna flóða. Þau hafa orðið að minnsta kosti tólf manns að bana. Hæsta viðbúnaðarstigi hefur verið lýst yfir í héraðinu. Ríkisfjölmiðlar í Kína hafa eftir Xi Jinping að ástandið í Henan sé afar alvarlegt. Stíflur hafi brostið og valdið manntjóni og eignatapi. Allir verði að leggjast á árarnar til að koma í veg fyrir að það verði enn meira. Á annan tug borga og bæja eru umflotin vatni. Á götum hafa myndast straumharðar ár sem bera með sér bíla og alls kyns brak. Ef marka má fréttir af svæðinu er ástandið verst í héraðshöfuðborginni Zhengzhou. Þar flæddi vatn inn í jarðlestagöng með þeim afleiðingum að tólf drukknuðu. Um fimm hundruð var bjargað úr göngunum. Hátt í tvö hundruð þúsund íbúum borgarinnar hefur verið forðað að heiman vegna flóða. Síðustu þrjá sólarhringa hefur fallið álíka mikið regn og á einu ári. Þá greindi kínverski herinn frá því í gær að tuttugu metra sprunga væri komin í Yihetan stífluna í Luoyang þar sem um það bil sjö milljónir búa. Hermenn hafa verið sendir á vettvang til að stýra rennsli í ám og hlaða upp sandpokum til að styrkja bakkana. Vegna veðursins í miðhluta Kína hefur fjölda flugferða verið aflýst og áætlunarferðir járnbrautarlesta eru úr skorðum. (e. Around two hundred thousand residents of the city of Zhengzhou in Henan province in China have been evacuated from their homes due to floods. The floods have killed at least twelve people. The highest preparedness level has been declared in the province. State media in China have quoted Xi Jinping as saying that the situation in Henan is very serious. Dams have burst, causing casualties and property damage. Everyone must do their part to prevent further damage. Between ten and twenty cities and towns are flooded with water. On the streets, strong currents have formed, carrying cars and various debris with them. Judging by news from the area, the situation is worst in the provincial capital of Zhengzhou. There, water flooded into subway tunnels, resulting in twelve people drowning. About five hundred people were rescued from the tunnels. Almost two hundred thousand inhabitants of the city have been forced from their homes due to the floods. In the past three days, as much rain has fallen as in an entire year. The Chinese army reported yesterday that a twenty-meter crack has appeared in the Yihetan dam in Luoyang, where around seven million people live. Soldiers have been sent to the site to control the flow of rivers and build sandbags to strengthen the banks. Due to the weather in central China, numerous flights have been canceled and scheduled train trips are disrupted.) |
| **Summary** | Hæsta viðbúnaðarstigi hefur verið lýst yfir í Henan héraði í Kína vegna flóða. Að minnsta kosti tólf manns hafa látist (e. The highest level of preparedness has been declared in Henan province in China due to floods. At least twelve people have died.). |

| Gullfaxi$_{BASE}$ (RRN) | | ROUGE F1 |
|---|---|---|
| **Title** | skjálfti í henan-héraði | 25.00/0.00/25.00 |
| **Intro** | að minnsta kosti fimm eru látnir eftir að öflugur jarðskjálfti reið yfir kína í gær. | 35.29/11.43/29.41 |
| | R / C / L : 1/0/1 *The text is relevant to the topic of a natural disaster in China but incorrectly refers to the cause of the disaster as an earthquake. The language used is natural and grammatically correct.* | |
| **Summary** | minnst níu hafa látist í miklum flóðum í kína frá því í gær. tugir þúsunda hafa orðið að yfirgefa heimili sín vegna flóða. | 35.00/14.29/30.00 |

| Gullfaxi$_{BASE}$ (CNN/DailyMail) | | ROUGE F1 |
|---|---|---|
| **Intro** | að minnsta kosti 12 manns hafa verið drepnir í henan héraði í henan-héraði í kína | 54.55/28.57/48.48 |
| | R / C / L : 0/0/0 *The text is not relevant to the topic, incorrectly capturing the subject of the story, asserting that people were murdered in China. The language used includes unnatural repetitions.* | |
| **Summary** | að minnsta kosti 12 manns hafa látist í henan héraði í henan-héraði í henan-héraði í kína | 64.52/50.00/64.52 |

| Gullfaxi$_{BASE}$ (CNN/DailyMail + RRN) | | ROUGE F1 |
|---|---|---|
| **Intro** | hátt í tvö þúsund íbúar í kína hafa orðið að yfirgefa heimili sín vegna flóða í henan-héraði í kína. | 45.71/31.58/40.00 |
| | R / C / L : 1/0/1 *The text is relevant but contains a factual error regarding the number of people who left their homes due to the flood. The language used is natural and grammatically correct.* | |
| **Summary** | hæsta viðbúnaðarstigi hefur verið lýst yfir í kína vegna flóða í henan-héraði í kína vegna flóða. þau hafa orðið að minnsta kosti tólf að bana í henan-héraði í kína. | 78.95/58.54/73.68 |

| mT5$_{BASE}$ (RRN) | | ROUGE F1 |
|---|---|---|
| **Title** | Hæsta viðbúnaðarstig lýst yfir í Henan | 18.18/0.00/18.18 |
| **Intro** | Síðustu þrjá sólarhringa hefur fallið ástandið verst í miðhluta Kína. Hátt í tvö hundruð þúsund íbúar borgarinnar hafa verið fluttir að heiman vegna flóða. | 37.21/4.55/32.56 |
| | R / C / L : 1/0/0 *The text is relevant but the first sentence is unclear and unnatural.* | |
| **Summary** | Hæsta viðbúnaðarstigi hefur verið lýst yfir í Henan-héraði í Kína vegna flóða. Þau hafa orðið að minnsta kosti tólf manns að bana. | 82.05/65.00/76.92 |

| mT5$_{BASE}$ (CCN/DailyMail) | | ROUGE F1 |
|---|---|---|
| **Intro** | Hátt í tvö hundruð þúsund íbúar borgarinnar Sheng-sjá Zhengzhou í Henan-héraði í Kína hafa verið fluttir að heiman vegna flóða | 42.11/20.00/36.84 |
| | R / C / L : 1/1/1 *The text meets all criteria but note that it copies the first sentence of the input text.* | |
| **Summary** | Hæsta viðbúnaðarstig hefur verið lýst yfir í Henan-héraði í Kína vegna flóða | 60.00/46.67/60.00 |

| mT5$_{BASE}$ (CCN/DailyMail + RRN) | | ROUGE F1 |
|---|---|---|
| **Intro** | Flóðin í Henan-héraði í Kína hafa orðið að minnsta kosti tólf að bana. Stíflur hafa brostið og valdið manntjóni og eignatapi. | 54.05/43.90/54.05 |
| | R / C / L : 1/1/1 *The text meets all criteria. The first sentence effectively rephrases information from the input text while the second sentence is a direct copy.* | |
| **Summary** | Hæsta viðbúnaðarstigi hefur verið lýst yfir í Henan-héraði í Kína vegna flóða. Þau hafa orðið að minnsta kosti tólf manns að bana. | 82.05/65.00/76.92. |

Table 7: Example of model output. English translations are shown for the source text and the human evaluation scores are shown for each translation along with a justification.

| RRN Document | |
|---|---|
| **Title** | Stríðið gæti varað árum saman |
| **Intro** | Stríðið í Úkraínu gæti varað árum saman og Vesturlönd þurfa að vera undir það búin að veita Úkraínu stuðning áfram. Þetta segir Jens Stoltenberg framkvæmdastjóri Atlantshafsbandalagsins. |
| **Main** | Stoltenberg lét þessi orð falla í viðtalið við þýska dagblaðið Bild. Hann segir að þrátt fyrir að stríðið sé kostnaðarsamt verði Vesturlönd að halda áfram að senda Úkraínumönnum hergögn og nauðsynjar því kostnaðurinn verði umtalsvert meiri ef Vladimír Pútín Rússlandsforseti nái sínum markmiðum. Við þurfum að búa okkur undir það að stríðið gæti varað árum saman, segir Stoltenberg. Það sama segir Boris Johnson, forsætisráðherra Bretlands, sem fór í sína aðra heimsókn til Kænugarðs á föstudag. "It would be a catastrophe if Putin won. It would be a catastrophe if he was able to secure the land bridge to the cities in the south that he has, to hold the Donbas. That's what he wants. Boris Johnson segir skelfilegt að hugsa til þess að Pútín vinni stríðið. Stuðningur við Úkraínu sé nauðsynlegur til að koma í veg fyrir að Rússar nái Donbas. Breska varnarmálaráðuneytið birti í morgun yfirferð um átökin sem geisa hvað harðast við borgina Sjevjerodonetsk. Þar kemur fram að litlar breytingar hafi orðið síðasta sólarhringinn. Síðustu daga hafi verið nokkuð um liðhlaup úr úkraínskum hersveitum. Einnig segir breska varnarmálaráðuneytið að átök haldi áfram innan hersveita Rússa og dæmi séu um að hersveitir neiti að hlýða skipunum foringja sinna. |
| **Summary** | Framkvæmdastjóri Atlandshafsbandalagsins segir að Vesturlönd þurfi að vera undir það búin að veita Úkraínu stuðning áfram, stríðið geti varað í mörg ár í viðbót. |

| Gullfaxi_BASE (RRN) | | ROUGE F1 |
|---|---|---|
| **Title** | jens stoltenberg um úkraínu | 0.00/0.00/0.00 |
| **Intro** | vladimír pútín rússlandsforseti segir að stríðið í úkraínu sé í fullum gangi. pútín rússlandsforseti segir að rússar verði að sætta sig við stríðið í úkraínu. | 25.00/9.09/25.00 |
| **Summary** | framkvæmdastjóri atlantshafsbandalagsins segir að stríðið í úkraínu sé hafið. framkvæmdastjóri atlantshafsbandalagsins segir að stríðið í úkraínu sé hafið. | 40.00/6.25/33.33 |

| Gullfaxi_BASE (CNN/DailyMail) | | ROUGE F1 |
|---|---|---|
| **Intro** | forsætisráðherra bretlands segir að stríðið í úkraínu sé nauðsynlegurnauðsynlegur til að koma í veg fyrir að rússland nái árangri í úkraínu | 25.00/9.09/25.00 |
| **Summary** | jens stoltenberg, framkvæmdastjóri atlantshafsbandalagsins í úkraínu, segir að vladimírs pútín rússlandsforseti muni halda áfram að senda hergögn til kænugarðs. forsætisráðherra bretlands, donbas, segir að rússneskir hermenn hafi verið sendir til kænugarðs til að koma í veg fyrir átök í úkraínu. | 19.23/3.33/15.38 |

| Gullfaxi_BASE (CNN/DailyMail + RRN) | | ROUGE F1 |
|---|---|---|
| **Intro** | forsætisráðherra bretlands segir skelfilegt að stríðið í úkraínu sé að undirbúa sig fyrir það. hann segir að stríð í úkraínu sé á réttri leið. | 28.57/8.70/28.57 |
| **Summary** | jens stoltenberg knattspyrnustjóri atlantshafsbandalagsins segir að stríðið í úkraínu gæti komið í veg fyrir að rússar nái friði í úkraínu. | 26.32/4.76/21.05 |

| mT5_BASE (RRN) | | ROUGE F1 |
|---|---|---|
| **Title** | Stríðið í Úkraínu | 25.00/0.00/25.00 |
| **Intro** | Forsætisráðherra Bretlands segir skelfilegt að hugsa til þess að Vladimír Pútín Rússlandsforseti nái sínum markmiðum. | 10.53/0.00/10.53 |
| **Summary** | Forsætisráðherra Bretlands segir að stríðið í Úkraínu gæti varað árum saman og Vesturlönd þurfa að vera undir það búin að veita Úkraínu stuðning áfram. Þetta segir framkvæmdastjóri Atlantshafsbandalagsins. | 62.22/36.00/44.44 |

| mT5_BASE (CCN/DailyMail) | | ROUGE F1 |
|---|---|---|
| **Intro** | Stuðningur við Úkraínu er nauðsynlegur til að koma í veg fyrir að Rússar nái Donbas segir Boris Johnson forsætisráðherra Bretlands | 18.60/0.00/13.95 |
| **Summary** | Stuðningur við Úkraínu er nauðsynlegur til að koma í veg fyrir að Rússar nái Donbas segir Jens Stoltenberg framkvæmdastjóri Atlantshafsbandalagsins | 25.00/0.00/10.00 |

| mT5_BASE (CCN/DailyMail + RRN) | | ROUGE F1 |
|---|---|---|
| **Intro** | Forsætisráðherra Bretlands segir skelfilegt að hugsa til þess að Pútín Rússlandsforseti vinni stríðið. Stuðningur við Úkraínu sé nauðsynlegur til að koma í veg fyrir að Rússar nái Donbas. | 20.83/0.00/12.50 |
| **Summary** | Stríðið í Úkraínu gæti varað árum saman og Vesturlönd þurfa að vera undir það búin að veita Úkraínu stuðning áfram, segir framkvæmdastjóri Atlantshafsbandalagsins. | 69.77/39.13/46.51. |

Table 8: Example of model output.

| RRN Document | |
|---|---|
| **Title** | Breytingar á leigumarkaði |
| **Intro** | Ungt fólk hefur hrakist af leigumarkaði í covid-faraldrinum og hefur í vaxandi mæli þurft að flytja aftur heim í foreldrahús. Vísbendingar eru um að dregið hafi úr framboði á leiguhúsnæði á síðustu mánuðum. |
| **Main** | Þetta kemur fram í könnun á vegum hagdeildar Húsnæðis- og mannvirkjastofnunar um stöðuna á leigumarkaði. Almennt hefur leigjendum fækkað á síðustu tveimur árum. Margir hafa nýtt sér hagstæð lánakjör til að kaupa húsnæði en aðrir hafa hrakist af markaðinum. Karlotta Halldórsdóttir hagfræðingur hjá stofnuninni segir að ungt fólk hafi orðið hvað verst úti. Svo myndi ég segja líka að áhrif af covid faraldrinum þau eru að koma einna verst niður á búsetu hjá ungu fólki. Yngsta aldurshópnum 18 til 24 ára. Þar erum við að sjá stórlækkun hjá þeim á leigumarkaði og þeir eru í auknum mæli að flytja aftur í foreldrahús Við mundum geta giskað á að það væri minni atvinna fyrir þau. Kannski mikið til fólk sem er í hlutastörfum, veitingageiranum, ferðaþjónustutengt og atvinna þar hafi minnkað og þau séu með minna ráðstöfunarfé og hafi þá ekki lengur efni á að vera á leigumarkaði Þá eru einnig vísbendingar um að dregið hafi úr framboði á leiguhúsnæði á undanförnum mánuðum. Í upphafi faraldursins jókst framboð verulega þegar Airbnb íbúðir fóru inn á hinn almenna leigumarkað. Við sáum einmitt í kjölfarið af covid og í fyrra þá vorum við að sjá framboð af leiguhúsnæði aukast töluvert. Fólk átti auðveldara með að finna sér húsnæði og svona aukið öryggi. En við erum að sjá í fyrsta skipti núna síðan 2015 að það sé hærra hlutfall sem finnst erfiðara að verða sér úti um húsnæði. Þannig að það eru vísbending um að það sé að draga úr þessu mikla framboði sem var Þetta hefur líka þær afleiðingar að leiguverð fer nú hækkandi. Við sáum það í fyrsta skipti í svolítinn tíma þá hækkaði vísitalan. alveg töluvert milli mánaða. þannig að það eru kannski svona einhver merki um að það sé aðeins farið að vera erfiðari leigumarkaðurinn hjá okkur |
| **Summary** | Staða ungs fólks á húsleigumarkaði hefur versnað í Covid faraldrinum og margir hafa neyðst til að flytja aftur í foreldrahús. Þetta sýnir ný könnun. |

| Gullfaxi_BASE (RRN) | | ROUGE F1 |
|---|---|---|
| **Title** | skortur á leiguhúsnæði | 33.33/0.00/33.33 |
| **Intro** | ungtungt fólk á aldrinum 18 til átján ára hefur flutt lögheimili sitt á almennum leigumarkaði. þetta kemur fram í nýrri könnun. | 20.83/0.00/20.83 |
| **Summary** | ungt fólk hefur dregið úr framboði á leigumarkaði frá því í fyrra. ungt fólk er líklegra til að flytjast búferlum frá heimilum sínum. | 26.67/4.35/22.22 |

| Gullfaxi_BASE (CNN/DailyMail) | | ROUGE F1 |
|---|---|---|
| **Intro** | ungt fólk á aldrinum 18 til 24 ára hefur fækkað á undanförnum tveimur árum. ungt fólk á aldrinum 18 til 24 ára eru verst úti á leigumarkaði. | 26.67/3.92/22.22 |
| **Summary** | ungt fólk hefur dregið úr framboði á íbúðarhúsnæði á undanförnum mánuðum. á undanförnum mánuðum hefur fjöldi leigjenda dregið úr framboði á leiguhúsnæði á undanförnum mánuðum. | 10.81/0.00/10.81 |

| Gullfaxi_BASE (CNN/DailyMail + RRN) | | ROUGE F1 |
|---|---|---|
| **Intro** | ungt fólk á aldrinum 18 til 24 ára er í auknum mæli á leigumarkaði í fyrsta sinn í 20 ár. þetta kemur fram í nýrri könnun sem gerð var á vegum húsnæðis- og mannvirkjastofnunar sem unnin var á dögunum. | 23.73/5.80/20.34 |
| **Summary** | ungt fólk hefur hrakist af leigumarkaði í vaxandi mæli á síðustu mánuðum. vísbendingar eru um að ungt fólk hafi dregið úr framboði covid-faraldrinum. | 21.74/0.00/17.39 |

| mT5_BASE (RRN) | | ROUGE F1 |
|---|---|---|
| **Title** | Leigumarkaður | 0.00/0.00/0.00 |
| **Intro** | Forstjóri Húsnæðis- og mannvirkjastofnunar segir að ungt fólk hafi orðið hvað verst úti um húsnæði. Þetta kemur fram í könnun á vegum hagdeildar Húsnæðis- og mannvirkjastofnunar um stöðuna á leigumarkaði. | 33.96/3.39/18.87 |
| **Summary** | Formaður Húsnæðis- og mannvirkjastofnunar segir að ungt fólk hefur hrakist af leigumarkaði í covid-faraldrinum og hefur í vaxandi mæli þurft að flytja aftur heim í foreldrahús. Vísbendingar eru um að dregið hafi úr framboði á leiguhúsnæði á síðustu mánuðum. | 28.57/9.68/25.00 |

| mT5_BASE (CCN/DailyMail) | | ROUGE F1 |
|---|---|---|
| **Intro** | Almennt hefur leigjendum fækkað á undanförnum tveimur árum. Margir hafa nýtt sér hagstæð lánakjör til að kaupa húsnæði. En aðrir hafa hrakist af markaðinum. | 19.61/3.64/19.61 |
| **Summary** | Almennt hefur leigjendum fækkað á síðustu tveimur árum. Margir hafa nýtt sér hagstæð lánakjör til að kaupa húsnæði. En aðrir hafa hrakist af markaðinum. | 25.53/8.51/21.28 |

| mT5_BASE (CCN/DailyMail + RRN) | | ROUGE F1 |
|---|---|---|
| **Intro** | Ungt fólk er í auknum mæli að flytja aftur í foreldrahús. Þetta segir hagfræðingur hjá Húsnæðis- og mannvirk-jastofnun. Almennt hefur leigjendum fækkað á síðustu tveimur árum. | 45.28/17.54/45.28 |
| **Summary** | Ungt fólk hefur hrakist af leigumarkaði og hefur í vaxandi mæli þurft að flytja aftur heim í foreldrahús. Vís-bendingar eru um að dregið hafi úr framboði á leiguhúsnæði á síðustu mánuðum. | 34.62/10.91/30.77. |

Table 9: Example of model output.