# OpenReview forum: "Abstractive Text Summarization for Icelandic"
_NoDaLiDa/2023/Conference — NoDaLiDa 2023_

### Official Review · Reviewer_DCqd · 2023-03-07
**Solid contribution with thorough model exploration + novel and useful resource creation**

**Rating:** 8
**Confidence:** 4

**Review:**

## Summary
This paper explores for abstractive summarization models for the Icelandic language. With the lack of a suitable dataset for abstractive modeling (prior work being extractive) a new dataset (RRN) is introduced, collecting and manually curating 4k articles from the Icelandic national broadcasting service (RÚV). Each article is annotated with an article body, an introduction paragraph, a summary, a and title. The <body, intro> is deemed abstractive, while <body, summary> is extractive. Two base models are considered for this dataset:
- A PEGASUS model trained from scratch on ~5 milion Icelandic news articles
- mT5
The paper explores fine-finetuning the models on different data, including translations of the English datasets: CNN/DM and XSum. Automatic metrics along human evaluation show that fine-tuning mT5 in two iterations, first on CNNDM and then on RRN produces the best results, while the PEGASUS-based model produces worse.

## Review
This is a wellwell-written paper that explores reasonable methods to develop an abstractive summarizer for the Icelandic language. Creating a dataset, manually ensuring data quality, training pre-trained language models from scratch, and testing different data configurations this paper contributions with considerable groundwork, and a adds valuable insights to Icelandic NLP. Some work on summarization already exists for Icelandic, however, it focuses primarily on extractive summarization [1] and as such this paper's contributions are in developing an abstractive dataset and evaluating systems on said abstractive data - less on showcasing abstractive systems feasibility on Icelandic language. The paper could be improved making including running the proposed system on previously published extractive dataset to see how it fairs compares. All in all, I find paper work valuable, and I am convinced the community benefits from being shared with the NLP community.

## Comments
- For readers not proficient in Icelandic it would be nice to have English translations of the examples included in the paper
- One might reconsider the length of the  background and perhaps instead focus on describing in greater detail how the dataset was created
- The paper incorporates the CNN/DM and XSum dataset, but only uses CNN/DM in two phase training. Why?
- For human evaluation you ask annotators to assign scores to each sample. Does this make sense? I'd imagine that properties of relevance/correctness/language are more continuous in nature, where a Likert scale might be more suitable.
- mT5 is trained on translations tasks unlike, while the Gullfaxi (PEGASUS-based model) only sees Icelandic. Is it possible that mT5 has an advantage in this regard as it's able to transfer MT capabilities?

[1] https://aclanthology.org/2021.naacl-srw.2

**Paper Type:**

Long paper

---

### Official Review · Reviewer_rY7v · 2023-03-13
**This paper makes a significant contribution towards abstractive text summarization in Icelandic.**

**Rating:** 9
**Confidence:** 4

**Review:**

The paper makes three contributions:
1. An icelandic dataset for abstractive summarization is created.
2. Several abstractive summarization models are trained and evaluated on this dataset, using different pretrained models.
3. The authors also experiment with data augmentation using machine translated data and evaluate its effect on different models.

Overall, these efforts make a nice contribution towards abstractive summarization in Icelandic as well as learning more about abstractive summarization in general in languages other than English.

Pros:
* A new dataset for a generative task is created for a language other than English.
* The experiments are thorough, exploring various different dimensions that could be relevant for the performance of the task (different pre-trained models, ways of fine-tuning, size of the fine-tuning data, several summarization tasks).
* In addition to using automatic evaluation, the authors have also carried out a human evaluation.

Cons:
* The human evaluation was done using dichotomous ratings, which does not seem to be the most appropriate choice here, because it assumes that a summary is either totally good or totally bad in a particular aspect which is rarely true. Thus, it can be hard to properly interpret the human evaluation results.
* It seems that all models were trained just once and thus the variance caused by the random seed cannot be assessed. This is perhaps not a too big problem, considering that the differences between models are quite large. Presenting the results with a precision of two digits after the comma seems excessive though in these circumstances.
* The background section seems too unfocussed and containing even irrelevant material (referring to Word2vec in this context seems a strech).

Questions/suggestions:
* Why didn’t you also train the task Main —> Summary?
* Why did you use binary criteria in human evaluation instead of a Likert scale?
* The section 2 could be rewritten in a shorter and more concise manner as currently the overall goal of this section remains somewhat unclear.
* The section 3.4 is worded such as if you are proposing the ROUGE measures, which is definitely not true. Refer to the appropriate sources to explain ROUGE measures and make it clear that these definitions are not yours.
* In section 4.1 it is said that the mT5base performs so good because it tends to be extractive. Perhaps it would be informative to also add the ROUGE scores ofthe extractive oracle baselines, i.e., taking the Intro to be the summary for Tasks 1 and 2? This way one could see what this kind of baseline would mean in ROUGE scores.
* The sentence in lines 831-837 seems confusing to me, maybe it can be reworded more clearly.


**Paper Type:**

Long paper

---

### Official Review · Reviewer_wUgd · 2023-03-16
**First abstractive text summarization dataset for Icelandic. Gullfaxi lost to mT5.**

**Rating:** 7
**Confidence:** 3

**Review:**

The paper focuses on abstractive summarization for Icelandic. A dataset for Icelandic is collected that, for example, includes English datasets (RNN/DailyMail and XSum) that are translated to Icelandic using MT. They show that a model pre-trained and fine-tuned solely on Icelandic (Gullfaxi) underperforms compared to the larger mT5 model.

The paper is well written and easy to follow. The presented work follows a rather obvious pipeline for resource development for a low-resource language. The dataset they collect and publish will likely be of value to others.

 * Table 2 and 4 - I suggest bolding the scores in each column to make them easier to interpret.

 * When looking at the Language criteria in Table 4, it seems to be some issues related to using the translated datasets. You mention that a human inspection was done. It would be interesting to hear a bit more about the quality of the translated sentences.

 * It would be interesting to hear more about the authors thoughts to why there is a difference in the scores achieved by mT5 and Gullfaxi. This would be a nice starting point for others who plan to work with this dataset. I am quite surprised to see how much better the mT5 model is performing compared to the Gullfaxi model. When looking at the human evaluation of Correctness in Table 4, here the difference is consistently large. Is it understandable that this difference can to a large extent be attributed to mT5 having more parameters and having been exposed to (a lot) more pre-training. But I am wondering if there could be some other differences that could impact these results? Are there some noteworthy differences in the architectures (except sizes)? Do you believe that some specific (hyper)parameter tuning of Gullfaxi could have resulted in better performance?

Pros:
 * The dataset will be of value to others.

Cons:
 * Not much novelty to be found in the approach used.
 * Gullfaxi performance is surprising when compared to mT5.


**Paper Type:**

Long paper

---

### Decision · Program_Chairs · 2023-03-17

Accept